# Genetic Diversity of Grapevine Virus A in Three Australian Vineyards Using Amplicon High Throughput Sequencing (Amplicon-HTS)

**DOI:** 10.3390/v16010042

**Published:** 2023-12-27

**Authors:** Qi Wu, Wycliff M. Kinoti, Nuredin Habili, Stephen D. Tyerman, Amy Rinaldo, Fiona E. Constable

**Affiliations:** 1School of Agriculture, Food and Wine, University of Adelaide, Waite Precinct, PMB 1, Glen Osmond, SA 5064, Australiasteve.tyerman@adelaide.edu.au (S.D.T.); 2Australian Wine Research Institute, Wine Innovation Central Building, Hartley Grove crn Paratoo Road, Urrbrae, SA 5064, Australia; 3Agriculture Victoria Research, Department of Energy, Environment and Climate Action, AgriBio, Centre for AgriBioscience, 5 Ring Road, Bundoora, VIC 3083, Australia; 4School of Applied Systems Biology, La Trobe University, Bundoora, VIC 3086, Australia

**Keywords:** amplicon sequencing, MiSeq, grapevine virus A, Shiraz disease, intra-host diversity, median-joining haplotype networks, phylogenetic

## Abstract

Shiraz disease (SD) is one of the most destructive viral diseases of grapevines in Australia and is known to cause significant economic loss to local growers. Grapevine virus A (GVA) was reported to be the key pathogen associated with this disease. This study aimed to better understand the diversity of GVA variants both within and between individual SD and grapevine leafroll disease (LRD) affected grapevines located at vineyards in South Australia. Amplicon high throughput sequencing (Amplicon-HTS) combined with median-joining networks (MJNs) was used to analyze the variability in specific gene regions of GVA variants. Several GVA^II^ variant groups contain samples from both vineyards studied, suggesting that these GVA^II^ variants were from a common origin. Variant groups analyzed by MJNs using the overall data set denote that there may be a possible relationship between variant groups of GVA and the geographical location of the grapevines.

## 1. Introduction

Shiraz disease (SD) affecting *Vitis vinifera* var. Shiraz and Merlot grafted onto 101-14 rootstock was first reported from South Africa in 1985 [1]. This disease was first detected in Australia in 2001 [2] and has since been frequently reported by Australian grape growers as one of the most destructive virus diseases that has resulted in significant canopy decline and yield loss [3,4,5]. The cost of removing infected grapevines and replanting, considering the elapsed time ahead of the first profitable crop, is estimated to be around AU$12 million yr^−1^ in Australia [4]. SD-affected grapevines display restricted spring growth (RSG), late bud burst and delayed canopy growth early in the growing season, leaf reddening, delayed leaf fall, and rubbery un-lignified shoots in autumn as compared to the asymptomatic grapevines [3,4,6]. 

Grapevine virus A (GVA), the core pathogen with a strong association with SD [4], is a positive-sense single-stranded RNA virus and is assigned to the family *Betaflexiviridae* under the genus *Vitivirus* [7]. This virus consists of 5 open reading frames (ORFs). ORF1 to 5 encode a 194 kilodaltons (kDa) RNA-dependent RNA polymerase (RdRp) (ORF1), a 19 kDa hypothetical protein (ORF2), a 31 kDa movement protein (MP) (ORF3), a 21.5 kDa coat protein (CP) (ORF4), and a 10 kDa RNA-binding protein (RB gene; p10 protein) (ORF5), respectively [8,9]. The p10 gene identified as the suppressor of gene silencing [10] consists of two major units: a basic arginine-rich domain and a zinc-finger domain [11]. One of the early studies demonstrated that replacing cysteine residues with serine in the zinc-finger domain of GVA did not affect binding; however, when inoculated onto *Nicotiana benthamiana*, the mutated GVA infectious clone did not induce symptoms compared to the wildtype [11]. Another study also suggested the eighth amino acid counted from the N-terminus of the GVA RB gene is involved in symptom severity [12]. Although these two studies indicate the involvement of the RB protein in symptom expression in an experimental host, others concluded that in the specific GVA phylogenetic groups (phylogroup) II (GVA^II^) and III (GVA^III^), genes on the 3′ terminal part of the genome, including partial MP, entire CP and RB, were linked to the SD status in grapevines and symptom severity in *N. benthamiana* [13,14]. They suggest the GVA isolates from GVA^II^ are always associated with SD in grapevines and severe symptoms on *N. benthamiana*, whereas isolates belonging to GVA^III^ were consistently associated with asymptomatic grapevines or induce milder symptoms compared to GVA^II^ isolates [13]. Phylogroup I (GVA^I^) was not associated with SD [15].

During virus replication, mutants emerge due to error-prone replication mechanisms, recombination, and reassortment, resulting in a viral quasispecies population. This can lead to molecular divergence and genetic variations within the same virus population [16,17,18]. Virus variation within its population can lead to breaking the resistance of the host [19,20] and harnessing natural selection to adapt to the current environment [21]. However, most mutations are inconsequential or harmless and are abandoned during virus evolution [21,22,23]. Studying the evolutionary history and genetic diversity of a virus population can provide information on genomic epidemiology as well as a deep understanding of the epidemiologic model of how viral diseases emerge and spread. 

Before the onset of the high-throughput sequencing (HTS) technology, the traditional way of studying virus population, including variants, in an individual involved reverse transcription polymerase chain reaction (RT-PCR), amplification of the gene of interest, then cloning amplicons into vectors followed by Sanger sequencing. Analysing sufficient clones to detect variants within an individual is time-consuming and costly and may miss important but low-frequency variants [24,25]. Amplicon high-throughput sequencing (Amplicon-HTS), which is a targeted sequencing approach, clearly shows its advantages over Sanger sequencing when investigating the depth of diversity in viral populations. It does not require any cloning steps, and can identify thousands of variants in a single sample with the ability to detect low-frequency variants [24], which may be an indicator of emerging pathogens. Amplicon-HTS has been successfully utilized in medical drug selection to detect drug resistance mutations [26,27,28], virus epidemiology studies for tracking the transmission of a viral outbreak [24,29,30], cancer research [31,32,33,34], investigating microbial diversity in environmental samples [35,36,37,38,39], and in human diseases [40,41,42]. Recently, this approach has been used to study virus evolutionary history, intra-host and inter-host diversities in plant virology, e.g., endive necrotic mosaic virus (ENMV) [43] in various hosts, several *Ilarvirus* species in *Prunus* species [44,45,46], and four viruses in legumes [47]. 

Median-joining networks (MJNs), a method that was first published in 1999 [48], is now widely used to study biogeography and, most importantly, the evolutionary relationships within a virus population [45,49,50,51,52]. The potential parent of a virus can be identified, and the evolutionary history can be visualized based on the principle of the median-join algorithm by which variants that are closely related in the network are also genetically closely related, and the parent variants should have the highest number of connections to the descendant variants [48]. This algorithm is based on the genetic distance of nucleotide sequence, which is similar to one of the most popular methods, the neighbor-joining algorithm, for constructing phylogenetic trees of viruses [53,54]. 

This study used deep amplicon sequencing combined with phylogenetic analysis and MJNs analysis to explore the intra- and inter-host diversity of GVA variants, the possible relationships between variant groups of GVA, and the relative geographical location of SD and grapevine leafroll disease (LRD) affected grapevines.

## 2. Materials and Methods

### 2.1. Sample Selection

The Amplicon-HTS experiment was carried out on a total of 26 Shiraz grapevines from three South Australian vineyards: Langhorne Creek (LC 1 to 15, clone BVRC12 grafted on Chardonnay rootstock), Willunga (WIL 1 to 8, own-rooted BVRC12), and Barossa Valley (BV 1 to 3, own-rooted unknown heritage clone, planted around 1900). The grapevines tested positive to GVA by two RT-PCR assays which used the primer pairs, Ah587/Ac995 and H7038/C7273 for the detection of CP and RB genes, respectively, during sample selection [55,56] (Appendix A). The samples were collected in May 2018 from grapevines that displayed typical SD symptoms, showing both RSG at EL stage 15 (8 leaves separated) and leaf reddening observed from EL stage 35 (veraison) to leaf-fall from LC and WIL, and from grapevines that showed mild LRD symptoms of pinkish leaf reddening without rolling, which was previously described for the BV vineyard which tested positive for grapevine leafroll-associated virus 1 [15]. Appendix A provides basic information on tissue type, variety, clone, year of planting, and number of grapevines per symptom type of each vineyard. 

### 2.2. Nucleic Acid Extraction Method and RT-PCR Amplification

Three dormant canes from the three vineyards were randomly sampled across the whole canopy of each grapevine in May 2018. Total nucleic acid (TNA) extraction was performed using the method described by Wu et al. [15]. The presence of RNA in each sample was determined using the ribulose bisphosphate carboxylase oxygenase (RubiscoL) internal control primers previously described [57] (Appendix A). A region from each of the MP, CP and RB genes was amplified from the 26 samples in a 25 µL reaction containing 0.1 µL of ProtoScript^®^ II Reverse Transcriptase (New England Biolabs, Ipswich, MA, USA), 1.25 µL of each primer of 10 µM primers, 12.5 µL of Q5 master mix (contains Q5^®^ High-Fidelity DNA Polymerase, New England Biolabs, Ipswich, MA, USA) 8.9 µL of PCR-grade water (Thermo Fisher Scientific, Waltham, MA, USA), and 1 µL of the TNA samples. The RT-PCR reaction conditions were as follows: reverse transcription at 42 °C for 50 min and initial denaturation at 98 °C for 30 s, followed by 35 cycles of denaturation at 98 °C for 20 s, annealing at 56 °C for 30 s, and extension at 72 °C for 30 s, with a final extension at 72 °C for 5 min. The desired amplicon sizes of MP, CP, and RB were 404 base pairs (bp), 565 bp, and 236 bp, respectively. The primer sequences and annealing temperature for the RT-PCR of each set of primers can be found in Appendix A.

### 2.3. Amplicon Purification and Library Preparation

To ensure that the amplicons were successfully amplified, 10 µL of each RT-PCR product was run on 1.8% agarose gel electrophoresis and stained with SYBR Safe™ gel stain (Invitrogen, Waltham, MA, USA). The remaining 15 µL was transferred to a fresh tube for amplicon purification using Ampure XP beads (Beckman Coulter, Brea, CA, USA) following the manufacturer’s protocol. The purified amplicons were first adapter-ligated using the Illumina mpxPE1 and mpxPE2 adapters and then PCR-enriched using the in-house multiplex PE barcode primer as described previously [45]. The reaction conditions for the PCR enrichment were the same as the RT-PCR amplification described previously, but only 15 cycles were used. The PCR-enriched amplicons were purified again using Ampure beads. The final concentrations of the 78 purified libraries were determined using the TapeStation system (Agilent, Santa Clara, CA, USA) and Qubit fluorometer (Thermo Fisher Scientific, Waltham, MA, USA). Libraries were pooled using an equal amount of each sample at a final concentration of 4 nM. The pooled libraries were then sequenced on an Illumina MiSeq instrument with 300 bp paired-end reads.

### 2.4. Amplicon Raw Reads Trimming and Filtering

Raw reads with a quality score of under 20 and a length below 50 bp, as well as the Illumina adapter sequences, were removed using AdapterRemoval (v. 2.2.2) [58] (quality trimming). The remaining reads from both directions were paired and merged using the “collapse” command of AdapterRemoval. Sequence reads were converted from “fastq” to the “fasta” format using the “reformat” function of BBmaps (v. 35.85) [59].

Amplicon sequence reads from the MP, CP, and RB genes were filtered to the designed size criteria: 399–404 nucleotides (nt) for MP, 560–564 nt for CP, and 230–244 nt for RB using the “reformat” function of BBmaps (size trimming). The size criteria differed for the MP, CP, and RB genes and were determined based on a preliminary analysis of the read length distribution of each gene region. Trimming and merging of reads were individually conducted for each sample using the “readlength” function of BBmaps. By considering all 26 samples, size criteria covering at least 80% of reads in the read-length histogram were established. This filtering process is implemented to exclude reads that are excessively long or short.

Afterward, the filtered amplicon sequence reads were compared against the local GVA sequence database using the “blastn” command of BLAST+ to determine the orientation of GVA amplicon sequence reads and to filter out the unspecific reads. The local database of GVA genome sequences was built using the “makeblastdb” command using BLAST+ (v. 2.11.0) [60], as described by Wu et al. [15]. 

The Cutadapt (v. 3.4) software was then used to remove primer sequences from the amplicon sequence reads, implementing a mismatch-tolerant function for variations in primer binding sites using default parameters [61]. 

After primer trimming, reads were filtered again to the expected sizes of 364 bp, 524 bp, and 198 bp for MP, CP, and RB genes, respectively. Additionally, the extra 38 nt sequence from the intergenic region right before the CP gene was deleted from the original CP amplicon sequences using Cutadapt’s “cutadapt” command, resulting in a final amplicon size of 486 bp for the CP gene.

Prior to clustering, an examination of insertions and deletions was conducted. Duplicates in primer-trimmed reads were eliminated from each sample using the “rmdup” command of SeqKit (v.2.3.0) [62]. Subsequently, the remaining reads were aligned using Muscle (v.3.8.31) [63]. The resulting aligned fasta files underwent visual assessment specifically focused on identifying insertions and deletions.

### 2.5. Amplicon Reads Error Filtering and Clustering

To remove RT-PCR and amplicon sequencing error-associated reads [45], reads with a frequency greater than 10 (cut-off value) in each sample were grouped at 100 percent nt identity using Usearch’s (v. 11.0.667) “fastx uniques” command [64]. Nucleotide sequence clustering results in several hundreds of unique nt variants, each of which was labeled by the copy number (N) obtained for that variant. The unique variants were named Cluster 1 to N by the copy numbers from high to low; e.g., Cluster 1 was the unique variant with the highest copy number (largest variant cluster) in a given sample from a particular gene. The unique nt variants were translated into amino acid (aa) sequences using the CLC Genomics Workbench (v. 21.0.3; Qiagen, Aarhus, Denmark). Protein sequences of each sample were clustered at 100% aa similarity again using the same “fastx uniques” command of Usearch. 

To study intra- and inter-host diversity within and between the three vineyards, after clustering of unique variants of each gene from each sample, data from all 26 samples of each gene were combined and again clustered as described above to obtain the combined pool of unique variants for each gene with the Usearch at 100% nt or aa sequence similarity.

### 2.6. Phylogenetic Group Identification of Amplicon Variants

Phylogroups of all publicly available near-complete genome sequences of GVA, including 37 Australian isolates from the GenBank database, were previously identified using a neighbor-joining tree [15]. These complete or near-complete GVA sequences were implemented as the customized database of the BLAST+ by the “makeblastdb” command to identify phylogroups for all amplicon variants in this study. The phylogroup of amplicon variants from each sample was assigned to GVA^I^, GVA^II^, or GVA^III^ according to the best match with the highest nt identity when blasted against the database described above using the “blastn” function of the BLAST+. To ensure the reliability of this method, the BLAST+ method was validated as follows. Variants containing phylogroup information were sorted by nt identities and sampled every 24 rows. The sampled variants were combined with GenBank sequences of known phylogroups and analyzed using a neighbor-joining phylogenetic tree by MEGA (v. 7.0.26) with 1000 bootstrap replicates [65]. The phylogroups, identified by the neighbor-joining tree and by the BLAST+ method, were compared to ensure they provided identical results before being applied to the isolates being analyzed in this study. 

### 2.7. Intra-Host Diversity Based on Lowest Pairwise Identities

In this study, the lowest percentage pairwise nt identities between any of the two amplicon variants in each sample were utilized as a parameter of genetic diversity within a virus population in an individual grapevine sample based on our hypothesis that a higher pairwise identity (closer to 100%) within a population has fewer sequence variations, whereas samples with lower nt identities contain deeper intra-host genetic diversity. The pairwise nt identity and aa similarity between each two amplicon variants were determined using the command version of the sequence demarcation tool (SDT, v. Linux64) [66]. The nt or aa identities were sorted from lowest to highest to find the lowest identities within the virus population using the default Shell command “sort” on the output generated from the SDT. 

### 2.8. Intra-Host Genetic Diversity by Median-Joining Haplotype Networks (MJNs) within a Single Sample

As the unclustered files were too large for the construction of a MJNs tree for each gene, only the 20 unique nt variants with the largest number of amplicon sequences (Cluster 1 to 20) from each gene of each sample were used for generating the scaled sequencing files as input files. In this study, a variant cluster consists of variants clustered at 100% nt identity extracted from the sequence data using the Usearch nt sequence clustering method as described above. The 20 largest clusters were used because together, they represented more than 50% of the trimmed reads before clustering in the majority of samples. Three samples, each from a different site, were selected as representative GVA-infected grapevines for the MJNs analysis. These included WIL3 and LC1, which displayed SD, and BV1, which showed mild LRD symptoms. Individual MJNs were constructed for each gene of each sample using the median-joining algorithm [48] by the PopART software (v. 1.7) (http://popart.otago.ac.nz (accessed on 25 July 2022)); thus, a total of nine MJNs were constructed. GVA^I^, GVA^II^, and GVA^III^ variants from each gene were identified using the Blast method described above and colored by different colors in the MJNs. The scale of the figure, to represent relative population size, for each gene of each sample was determined by dividing the number of sequence copies of larger clusters 1–20 by the number of sequences in the smallest cluster 20.

### 2.9. Intra and Inter-Host Diversity by MJNs for Overall Data

To study the evolutionary relationship among the GVA variants, the top 20 variant clusters from all 26 samples, in total 520 variants of each CP, MP, and RB gene, were combined and analyzed by the MJNs method described above. In this analysis, each variant cluster may consist of identical sequences obtained from different locations.

## 3. Results 

### 3.1. Sequencing Reads of Amplicon-HTS

Basic statistical data, including the number of raw reads, the number of reads after trimming, and the copy numbers of the top 20 variant clusters of each sample after clustering, as well as the number of nonsense variants, are reported in Appendix A. After the quality, size, and primer trimming, and before clustering, the total number of reads for the MP, CP, and RB genes, when all 26 samples were combined, was 2.13, 2.02, and 2.81 million, respectively (Table 1). Blast analysis of the discarded reads, after quality and size trimming, against the GenBank database indicated that shorter reads were either GVA-associated or *V. vinifera* (host) associated. The average proportion of reads for each of MP, CP, and RB per sample after quality, size, and primer trimming was 64.44%, 46.08%, and 73.36% (primer trimmed reads/total raw reads) (Table 1). It was observed that after size trimming, the average proportion of the remaining reads for the MP gene of BV1, BV2, and BV3 was only 25.99%, 9.23%, and 2.44%, and the proportion of the CP gene reads for WIL7 was 17.84%, which was much lower than the average proportion of reads for these genes (Appendix A).

### 3.2. Nucleotide and Amino Acid Sequence Clustering

The nucleotide sequence of the amplicon reads, which passed through all three trimming filters, was clustered at 100% identity at nt and aa levels. The number of clustered unique nt and aa variants detected in each sample of each gene is listed in Table 1. After nucleotide sequence clustering, a total of 10,454, 10,899, and 6366 unique variants were discovered from MP, CP, and RB genes, respectively. The proportion of unique variants to total variants of the MP, CP, and RB sequences was 80.52%, 83.18%, and 63.33%, respectively (Table 1).

Each sample’s unique nucleotide variants were translated to aa sequence and grouped again at 100% aa similarity, resulting in a total of 6593, 5389, and 6192 protein variants for the MP, CP, and RB genes, respectively (Table 1). Among these, 4402, 2250, and 1900 are distinct protein variants, accounting for 66.77%, 41.75%, and 30.68% of total protein sequences of the MP, CP, and RB genes, respectively.

Unique nonsense aa variants with premature stop codons were found in each of the three genes in 76/78 samples. Nonsense variants were found in the CP and RB genes but not in the MP gene of samples BV2 and BV3 (Appendix A). Generally, they represented < 10% of the total unique aa variants, except for the CP gene of isolate BV2, where unique nonsense variants represented 33.02% of the total (35 unique nonsense aa variants/106 unique aa variants Appendix A).

### 3.3. Intra- and Inter-Host Diversity of Amplicon Variants

Appendix A provides the phylogroups of GVA variants in each sample. Variants from all three phylogroups were identified in the CP dataset, whereas only GVA^I^ and GVA^II^ variants were obtained from MP and RB genes. Within the MP data set, all variants from BV belonged to GVA^I^, and all variants from LC and WIL belonged to GVA^II^. For the CP data set, GVA^III^ variants were detected along with GVA^II^ in the same grapevine in three samples, WIL2, 3, and 8 (Appendix A).

#### 3.3.1. Intra-Host Diversity: Lowest Pairwise Identities within Each Sample

The largest genetic distance (the lowest nt identity) was found in the CP gene of WIL2, WIL3, and WIL8 when the GVA^II^ and GVA^III^ variants found within each of these samples were compared, ranging from 75.7–76.1% (Appendix A). Comparisons within GVA^II^ and GVA^III^ variant populations in these three samples showed the lowest genetic distance ranging from 90.7–91% and 88.1–91% nt identity, respectively. This was higher than the genetic distance observed for the CP gene within other samples, which ranged from 93.6–99.1% nt identity and 93–97.5% for GVA^II^ and GVA^I^ variants, respectively. The highest genetic distance (90.11% nt identity) for the MP gene was observed within GVA^II^ variants in sample LC8, while the overall range of intra-host diversity amongst the other 25 samples ranged from 90.4–98.9% and 96.2–96.4% nt identity for GVA^II^ and GVA^I^ variants, respectively. For the RB gene, the highest genetic distance (91.4% nt identity) was observed within GVA^I&II^ variants in samples LC3 and BV2; the overall range of intra-host diversity amongst the other 24 samples ranged from 93.9–97% nt identity.

#### 3.3.2. Inter-Host Diversity: Lowest Percentage Pairwise Nucleotide Identities and Amino Acid Similarities within Each Phylogroup

The MP, CP, and RB variants of GVA^I^, GVA^II^, and GVA^III^ were compared across all 26 samples, and the overall lowest pairwise nt identities and aa similarities for each gene within each phylogroup across all 26 samples are listed in Table 2. Note that the lowest % nt identity of each phylogroup of GVA was obtained based on an unequal number of samples of each phylogroup (Appendix A). The lowest % nt identities and aa similarities of GVA^I^, GVA^II^, and GVA^III^ were obtained from 3 (BV1 to BV3), 23 (whole data set excluding BV), and 3 (WIL2, WIL3 and WIL8) samples, respectively. The number of variants used to gain the % nt identities and aa similarities of each phylogroup of GVA and each gene are given in Appendix A. Based on the current data set, only the CP gene could be compared across all phylogroups. For this gene, the % nt identity was higher in GVA^I^ variants, indicating the lowest variability, but when the sequences were translated to aa, the % similarity was lower than that of GVA^II^ and GVA^III,^ indicating high variability (Table 2). The nt identities and aa similarities of the MP gene were higher for GVA^I^ compared to GVA^II^, indicating the lower variability of the GVA^II^ variants (Table 2). Since GVA^I^ and GVA^II^ variants cannot be segregated based on the RB gene [15], the lowest % nt identities and aa similarities of the RB gene were calculated using combined variants from both groups. Therefore, identities for the GVA^II^ RB gene were not shown in Table 2.

### 3.4. Intra- and Inter-Host Diversity Analysis by Median-Joining Haplotype Networks (MJNs)

#### 3.4.1. MJNs of Sample WIL3, BV1 and LC1

Intra-host diversity of GVA in three samples of grapevines, each from a different region and displayed SD (WIL3, LC1) or mild LRD (BV1), were visualized by MJNs of the nt sequences of the MP, CP, and RB genes (Figure 1). Each subfigure represents a given gene of a single sample. BV1 was chosen because it contains only GVA^I^ variants. Samples WIL3 (Figure 1a,b) and LC1 (Figure 1g,h) were chosen for analysis because they have the highest copy numbers of GVA^II^ variants for the CP and MP genes. WIL3 was also chosen because it contains GVA^III^ CP variants that occur with much lower frequency compared to GVA^II^ variants (Figure 1a). 

GVA^II^ variants were widely segregated from GVA^III^ variants of the CP gene in sample WIL3, as indicated by the large number of hatch marks (Figure 1a), which represent the number of mutation events separating the variants and, therefore, evolutionarily distantly related. For the CP, MP, and RB genes of the GVA^II^ variants in WIL3, there was one large cluster (Cluster 1) surrounded by and connected to smaller clusters, and the smaller clusters were not connected, suggesting that Cluster 1 of each gene is likely the ancestor of other variants in this sample (Figure 1a–c). The number of copies in both Cluster 1 of the CP and the MP represented approximately 75% of the total number of copies amongst all clusters, whereas RB Cluster 1 represented 92% of all copies because it contains both GVA^I^ and GVA^II^ variants.

GVA^I^ variants of the CP gene from sample BV1 showed dispersed distribution, and no major variant was found (Figure 1d). However, Cluster 8, which has 983 copies, has the highest number of connections to other clusters, suggesting it may be a primary ancestor. Only one major cluster was found among GVA^I^ MP variants. As seen from Figure 1e, four mutation events gave rise to GVA^I^ MP Clusters 2, 15, and 14, and then four and then two mutation events gave rise to Clusters 17 and 20, which are most distantly related to Cluster 1. The GVA^I^ variants of the RB gene, Cluster 2, have the highest number of links to other clusters, and although Cluster 1 and Cluster 3 appear to give rise to some other clusters, they are closely related to Cluster 2 as they are only separated by one hatch mark. Therefore, Cluster 2 is the most likely ancestor of this population (Figure 1f).

Two major variants were found for the LC1 GVA^II^ MP (Figure 1h) and RB genes (Figure 1i), whereas three were found for the CP gene (Figure 1g). In a previous experiment using metagenomic high throughput sequencing (Meta-HTS) of sample LC1, two distinct molecular GVA^II^ variants were found, namely LC1-1 (accession no. MT070963) and LC1-2 (accession no MT070962), which share 92.90% nt identity by pairwise alignment of their genome sequences using Blast [60]. In the context of this study, the percentage of pairwise nt identities of the largest five clusters of CP and MP genes show that the top 5 clusters have over 99.4% nt identity to the isolate LC1-1 or to LC1-2 (Table 3). The CP gene Cluster 3 and MP gene Cluster 1 were identical to isolate LC1-1 and CP gene Cluster 1, respectively, and MP gene Cluster 3 were identical to LC1-2, supporting the detection of distinct GVA^II^ variants in a single sample (Table 3). The nt identity of the same region of the CP (486 nt) and MP (364 nt) between LC1-1 and LC1-2 are 96.09% and 93.96%, respectively. CP Cluster 2 is the primary ancestor of sample LC since it is surrounded by Clusters 1, 4, 6, 19, 12, 15, and 18 (Figure 1g). Clusters 2 and 5 are distantly related (97.33% nt identity) compared to CP Clusters 3 and 5 (98.97% nt identity) (Figure 1g). CP Cluster 3 and MP Cluster 1 are smaller and give rise to fewer variants than CP Cluster 1 and MP Cluster 3, respectively (Figure 1g,h). Cluster 1 of the RB gene is identical to LC1-1, but the similarity between clusters and LC1-2 could not be calculated as the RB gene sequence of MT070962 is incomplete (Table 3). However, two distinct RB gene clusters were formed, which could be representative of the LC1-1 and LC1-2 (Figure 1h,i).

#### 3.4.2. Intra- and Inter-Host Diversity of MJNs by the Overall Data Set

After sequence clustering, the top 20 variant clusters (one sequence per cluster) of each gene from each of the 26 grapevines were combined to draw an overall MJNs map for each gene (Figure 2). In these MJNs, the size of the variant cluster displays the number of samples within a given cluster (100% nt identity), and the colors display sample locations. When the MJNs were constructed, variants were clustered together and formed into variant groups. There were 17, 14, and 10 variant groups formed in the MJNs for the CP, MP, and RB genes for GVA^II^, respectively (Table 4). For each gene, some variant groups were only represented by variants from a single grapevine (single-sample variant group), but some were represented by variants from multiple grapevines (multiple-sample variant group). Variant groups were each labeled with a variant group ID (VG1 to VG21), and the lowest % nt identities of variants within each variant group are given in Table 4. The multi-sample variant group 14 (VG14) of the RB gene contained GVA^II^ variants of ten grapevines with a total of 546,940 reads, which count as 19.43% of the total amplicon reads for this gene after primer trimming (Table 4 and Appendix A). GVA^II^ variants associated with grapevines located at LC (LC1, LC2) and WIL (WIL1) occurred in muti-sample variant group VG4 across all three genes (Table 4; Figure 2a), and the MP, CP, and RB genes of the GVA^II^ variants across the three samples shared 38, 141, and 208 identical variants, respectively. 

GVA^II^ variant groups radiate from multi-sample variant group VG4 (grapevines LC1, LC2, WIL1) based on the CP and MP genes, suggesting that variants within these CP and MP variant groups are likely to be the ancestor of all other GVA^II^ variants (Figure 2a,b; Appendix A; marked with green dots). However, the ancestral relationship was not supported by the RB gene since variant groups appear to arise from multi-sample variant group VG14 that contains variants of WIL1 along with variants of nine other grapevines but not those from LC1 and LC2 (Figure 2c). For the CP gene, GVA^I^ and GVA^III^ variants were not connected to each other. Instead, they were only connected to GVA^II^. However, they were separated from GVA^II^ by a large number of hatch marks, suggesting a distant genetic relationship (Figure 2a). 

Based on current data of three samples, BV1, BV2, and BV3, GVA^I^ variants formed a diverse and interconnected group of unique variants across all three genes. Based on the size and number of linkages to other clusters of the CP and RB genes, the potential ancestor variant of the three BV samples was labeled by red circles in Appendix A. The evolutionary relations within the MP gene could not be determined (Appendix A), likely due to the low number of reads that were obtained and analyzed. 

#### 3.4.3. Variant Groups vs. Geographical Location

The relative geographical locations of the SD grapevines and their GVA CP gene variant groups were linked together in Figure 3. Generally, samples within the same GVA variant groups are geographically close to each other and are most frequently next to each other in a vineyard. There are exceptions, e.g., grapevines LC5 and LC13 are both within GVA CP variant group VG3, but they were physically distant within the vineyard (Figure 2a and Figure 3a). In contrast, WIL1 is next to WIL2 and WIL3, but its GVA^II^ variants belong to variant group VG4, which also contains GVA^II^ variants from grapevines LC1 and LC2 (Figure 2b and Figure 3b). Grapevines WIL4, 5, and 6 are neighbors occurring in a row but GVA^II^ CP, MP, and RB variants WIL5 and 6 occur in variant group VG8, and GVA^II^ variants in WIL4 occur in variant group VG18 (Figure 2 and Figure 3b). 

## 4. Discussion

This study used Amplicon-HTS of the MP, CP, and RB genes of GVA to demonstrate the complexity of the genetic structure of a population of GVA variants in samples of individual grapevines expressing either SD or mild LRD symptoms in different vineyards. In this study, GVA variants refer to genetic strains that coexisted within a GVA population. Plant virus populations can change with time in specific tissues due to genetic drift or selection [67,68], but this was not measured in this study. Instead, a single pooled sample (3 canes per grapevine) was collected from each of 26 grapevines across three different vineyards to provide a snapshot of GVA diversity in individual grapevines, within individual vineyards, and between vineyards at a single point in time. Even with this limitation, GVA diversity was observed and demonstrated the potential emergence of new variants within a population, but it also demonstrated that most variants within an individual grapevine are likely to be related. This study also demonstrated relatedness between GVA populations in some grapevines within and between vineyards, which is likely to be associated with a combination of insect transmission and spread via propagation material. 

### 4.1. Challenges of GVA^I^ Amplification by the MP Primers

The number of reads in the BV samples (BV1 to BV3) was significantly reduced post-filtering, as indicated in Appendix A. In fact, the unfiltered reads from BV contain a substantially higher percentage of host-associated reads compared to samples from WIL and LC, which may indicate the binding specificity of the MP primer to GVA^I^ variants was low. At the time of this study, the specificity of the MP primers for the detection of GVA^I^ strains could not be determined as sequence information was unavailable. Subsequent analysis has identified that there are four nt between the forward primer and the complete genome sequences of GVA^I^ from BV (accession no. OP752632). The shorter GVA-associated sequences were likely a result of incomplete PCR amplification, where some amplicons were only partially amplified or due to the occurrence of incomplete segments during the sequencing process.

### 4.2. Lowest Pairwise Identities

Diversity was found in all genes when assessing the largest genetic distance between GVA variants using lowest nt/aa identities and maximum nt/aa changes. It has been noted that the genetic distance may differ between gene regions due to mutation [69,70,71]; for each GVA phylogroup, a similar depth of diversity was observed for each gene within individual samples (intra-host diversity) as was observed between samples (inter-host diversity). The largest genetic distance (up to ~12% and ~13.5% intra- and inter-host nucleotide diversity, respectively) was observed in the CP gene; however, high levels of diversity were observed for the MP gene (up to ~9.6% and ~11.3% intra- and inter-host nucleotide diversity, respectively) and RB gene (up to ~9.9% and ~13.6% intra- and inter-host nucleotide diversity, respectively) in some samples. Amplicons of different sizes can arise due to incomplete extension, insertions, deletions, and chimeras during amplification [72,73], leading to the inadvertent identification of artificial variants that affect downstream analyses. Therefore, untrimmed and trimmed sequences were filtered to the same size to ensure the accuracy of the downstream phylogenetic and intra-host diversity analyses. Additionally, prior to further analysis, the trimmed sequences of the correct length were examined for the presence of insertions and deletions, and none were identified. As a result, it is reasonable to conclude that the observed diversity of GVA within and between grapevines is most likely attributed to naturally occurring mutation events, even though this study did not quantify or measure them. 

RNA viruses naturally have high mutation rates [21], which could lead to the high level of diversity observed in this study. Previous studies have suggested the mutation rate of RNA viruses can be different depending on the virus species as well as the selection pressure imposed by the plant host, vector species, and environment [21,74], which results in evolution. It is possible that these factors have contributed to the high level of diversity of GVA observed in many of the grapevines. However, not all samples demonstrated a high level of diversity, and the lowest level of diversity was ~1% for each of the three genes. The differences in the amount of diversity observed amongst the 26 grapevines could be associated with differences in selection pressures within and between the different vineyards. In potatoes, it was found that vegetative propagation led to higher diversity, while insect transmission reduced diversity [75]. 

As depicted in Table 1, the translation of nt to aa sequences, followed by grouping at 100% aa similarity, revealed a notable reduction in the number of unique aa variants compared to nt variants. This observation suggests a predominant occurrence of synonymous substitutions among nucleotide changes. Although synonymous substitutions conventionally are thought to have no direct impact on protein structure or function, studies have demonstrated their role in changing virus fitness by altering the secondary structure of RNAs and modifying codon usage bias during replication [76,77,78]. These alterations may lead to a reduction in translation and replication efficiency, resulting in decreased infectivity during transmission and weakened protein expression during virus evolution [77,78,79]. The intricate interplay between synonymous and nonsynonymous mutations in the GVA population under natural selection pressure and their implications for pathogenicity and the evolution of GVA merits further exploration.

It is worth noting that GVA^I^ variants in this study had greater variability at the protein level compared to the nucleotide level, with an 86.42% aa similarity versus a 92.59% nt identity for the CP gene (Table 2). This is somewhat unusual considering the crucial role of the CP protein in virus and host or vector interactions. Changes in amino acids can impact the coat protein structure and potentially affect interactions between hosts, vectors, and the virus [80]. Previous studies have demonstrated that RNA viruses with faster replication rates tend to have higher mutation rates [81]. The GVA^I^ variants examined in this study were obtained from a heritage Shiraz vineyard established in the early 1900s. In this particular case, the observed variability of the GVA^I^ variants may not have a significant impact on long-term virus–host interactions, or there may be a low selection pressure at this specific site.

### 4.3. Intra-Host Diversity

This part of the study has been used to explore the evolutionary relationships between each of the distinct variant clusters as well as the population size of each cluster within three representative samples, each from a different location. The MJNs method used in this study detects single nucleotide variations within GVA populations, which provides an in-depth visualization of the population diversity. This study showed that for each of the CP, MP, and RB genes, a GVA^II^ population within a host has at least one major variant cluster. Numerous unique variants of low copy number that are connected but diverge from major clusters occur in a star-burst pattern. The major cluster usually represents the majority variant of the virus population in each grapevine, such as that observed for sample WIL3. This could represent a single, possibly recent, infection event followed by the evolution and emergence of new strains. However, in some individual grapevines, there was evidence of two or more distinct large copy number clusters of GVA^I^ and GVA^II^ variants in addition to the smaller low copy number variants, such as that observed with grapevine samples BV1 (GVA^I^) and LC1 (GVA^II^). GVA^I^ variants display a more dispersed distribution than GVA^II^ with no major variant cluster and with similar cluster size. In these instances, there are some possible explanations: multiple infections have occurred either at the same time or at different times; a longer-term infection has given rise to multiple successful GVA variant clusters from a primary cluster through many generations; or the three pooled canes were infected with one or more different variant clusters.

Both GVA^II^ and GVA^III^ variants were detected in three samples (WIL2, WIL3, and WIL8) using Amplicon-HTS, although the presence of GVA^III^ was indicated only by the detection of the CP gene. This finding partially aligns with the results of our previous investigation [15], which used metagenomic sequencing and phylogenetic analysis of complete genomes to investigate GVA diversity in the same grapevines. In that study, mixed infections of GVA^II^ and GVA^III^ were detected in the same three samples as well as in WIL1. None of these mixed GVA^II^ and GVA^III^ isolates showed evidence of recombination [15]. Therefore, both studies provide evidence that the genetic variation in the CP gene is likely a result of the presence of two distinct molecular variants within the same grapevine rather than being attributable to recombination events [15]. The absence of GVA^III^ detection in the MP and RB genes by PCR and amplicon sequencing may be attributed to inadequate amplification, which could result from suboptimal primer specificity, a low GVA^III^ titer, or a combination of both factors. Nevertheless, the detection of both phylogroups based on the CP gene supports the notion of multiple GVA strain infections, possibly stemming from distinct infection events. To further explore the evolutionary pathway of GVA^III^ variants, Amplicon-HTS should be employed, using either group III variant-specific primers or universal GVA primers.

GVA^II^ variants are often associated with SD [15]. When this study was conceptualized, the possibility that a dominant SD-associated GVA^II^ variant might be observed amongst the SD-affected grapevines was considered. However, no single dominant variant was observed, and further study is required to understand whether any linkage between the variant characteristics of GVA and SD exists.

Nonsense mutations naturally exist at a very low level of frequency, and the population of nonsense mutants may be reduced during virus evolution, possibly by the nonsense-mediated mRNA decay in the host [82,83]. Little is known about the impact of nonsense mutations within a virus population; however, it is known that nonsense mutations of the RdRp and MP gene attenuate the symptoms of tomato mosaic virus [84] and nonsense mutation affect virus transmissibility of a SARS-CoV-2 strain [85]. In this study, nonsense variants were found in 76/78 sequencing samples, but they were mostly low-frequency variants and were not included in the analysis as they did not fall within the 20 largest variant clusters within any sample. Future research should investigate which of these nonsense variants were naturally occurring and which were generated through PCR error. Then, the effect of naturally occurring nonsense mutation on the severity of symptoms and transmission of GVA should be investigated. 

### 4.4. Evolutionary Relationship between GVA Phylogroups

MJNs of intra- and inter-host diversity provide a snapshot of the evolutionary relationship between GVA phylogroups. Based on the CP gene, GVA^I^ and GVA^III^, which do not have an association with SD [15], are more distant from each other than they are from GVA^II^ and are linked to GVA^II^ by a large number of hatch marks (Figure 3), indicating many mutational events have occurred during their evolution. The direction of evolution of a virus towards milder or more virulent strains can be difficult to predict since many complicated factors, such as virus and host interaction, environment adaptation, and vector transmission, lead to evolution [86,87,88]. However, mild virus strains, including GVA, can go undetected if symptoms are not obvious, resulting in the selection and distribution of infected symptomless or asymptomatic material by propagators and longer-term survival in the field. As a consequence, milder GVA strains may achieve a mutualistic or commensal symbiotic relationship with their hosts [89] and could have contributed to the emergence of GVA^I^ and GVA^III^. Evidence for this is that virulent strains were overtaken by the mild strain of Omicron during the SARS-CoV-2 global pandemic [90]. To determine whether GVA^II^ variants represent the ancestral phylogroup, diversity characterization of the most variable samples should be carried out using a complete genome sequence. 

### 4.5. Origin of GVA Variants in Vineyards

Two major modes of GVA transmission can occur either via infected propagation material or transmission by insect vectors (mealybugs and scales) [91,92,93]. The haplotype analysis of this study demonstrates a common origin of GVA^II^ variants within a vineyard, which is likely to be associated with spread via an insect vector. The first and second-instar nymphs are most efficient at transmitting viruses [92,94,95,96], which occurs as they crawl from vine to vine. This could explain the infection of adjacent grapevines at the LC site. Insects can also be carried on people’s clothing and machinery and can be dispersed by wind or water over short distances [97,98].

An alternate possibility for adjacent grapevines to become infected with the same variant group is the use of infected planting material or topworking grapevines already infected with the same GVA variant groups originating from the same source. The first piece of evidence to support this hypothesis is the presence of the same CP, MP, and RB variant groups in grapevines at LC (LC1, LC2) and WIL (WIL1) vineyards that are approximately 70 km apart. While the long-distance movement of vectors by machinery is known to occur, which cannot be completely discounted [99], LC and WIL are not managed by the same growers, and sharing of equipment is unlikely. Therefore, the distance between vineyards would make insect spread of GVA between these two locations unlikely. The second piece of evidence for spread is vegetative propagation, in which different sources of propagating material are used. For example, WIL4 was infected with a different variant group compared to WIL5 and WIL6, yet all three grapevines occurred in the same row. 

Alternatively, it is possible that a proportion of GVA variants may have been selectively passed from the original source into a new host due to bottleneck events occurring in the vector [100,101] or by escaping plant defense and adapting to a new host [21]. More evidence supporting the selective transmission of GVA variants is that Shiraz WIL9 and WIL10 had only GVA^III^ variants, while WIL8, which was next to WIL9, had both GVA^II^ and GVA^III^ variants.

## 5. Conclusions

This study has demonstrated the effectiveness of deep Amplicon-HTS as a valuable tool for investigating the epidemiology of SD, including its transmission and the evolutionary pathway of GVA. It has provided a deeper understanding of the diversity of GVA within both SD- and LRD-affected vineyards, as well as within individual infected grapevines. The MJN analysis has successfully linked variant groups to the geographical locations of the grapevines, thereby illustrating the expected pathway of GVA transmission to grapevines through insect vectors and plant propagation, ultimately leading to the spread of SD. Future analyses should include a thorough investigation of low-frequency variants to elucidate their contribution to the evolutionary history of these viruses. To gain a deeper insight into the epidemiology of GVA and SD, it is essential to conduct a rigorous analysis utilizing complete genome sequences and to compare these findings with the results presented in this study. This approach should encompass a larger dataset, incorporating both SD-affected and unaffected grapevines co-planted within a vineyard, as well as several SD-affected vineyards planted with the same grapevine clone. Furthermore, delving into the pathogenic traits of individual GVA variants at both nucleotide and protein levels is essential. Investigating the correlation between nt and aa changes and understanding how these aa changes contribute to modifications in protein functional motifs necessitates the use of infectious clones to provide a comprehensive understanding of the association between each GVA variant and their role in the symptom expression of SD.

## Figures and Tables

**Figure 1 viruses-16-00042-f001:**
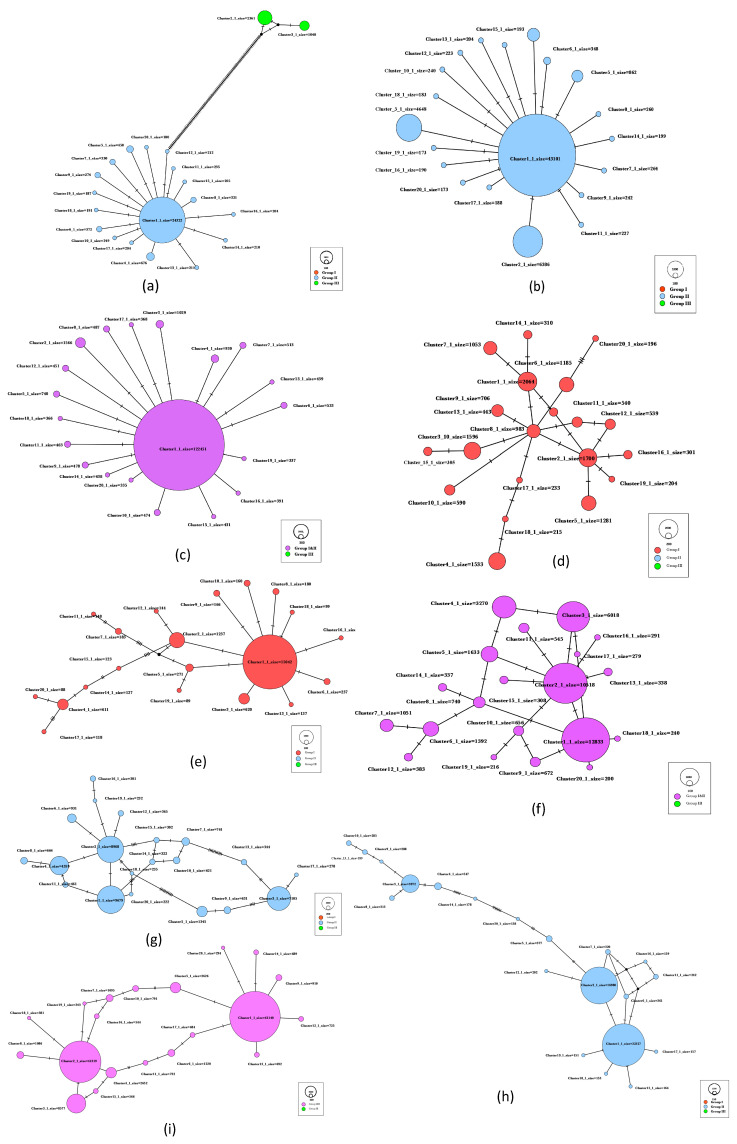
The intra-host diversity of coat protein (CP), movement protein (MP), and RNA-binding (RB) gene regions of grapevine virus A (GVA) in three grapevine samples, presented as median-joining haplotype networks (MJNs). (**a**–**c**), MJNs of the CP, MP, and RB gene, respectively, of the sample WIL3 which showed typical Shiraz disease symptoms from the Willunga site. (**d**–**f**), MJNs of the CP, MP, and RB genes, respectively, of the sample BV1, which showed mild leafroll disease symptoms from Barossa Valley. (**g**–**i**), MJNs of the CP, MP, and RB genes, respectively, of the sample LC1, which showed typical SD symptoms for Langhorne Creek. Each of the red, blue, and green colors indicates the phylogenetic groups of each variant in each sample. GVA variants from phylogenetic groups I and II (GVA^I^ and GVA^II^) of the RB gene were unable to be segregated due to the high similarity of this gene [15] and labeled using purple named as GVA^I&II^. The size of the circle represents the copy numbers of the amplicon variant that it represents. The scale of each subfigure was individually assessed based on copy numbers of Cluster 20 in each data set. The 20 variant clusters with the largest number of amplicon sequences (Clusters 1 to 20) from each gene were chosen for analysis. The original copy numbers of each variant are given by “size=”. The hatch mark indicates the number of mutations between each of the two variant clusters. The hypothetical median vectors (black dots) were implemented when the connection of two variant clusters was unknown.

**Figure 2 viruses-16-00042-f002:**
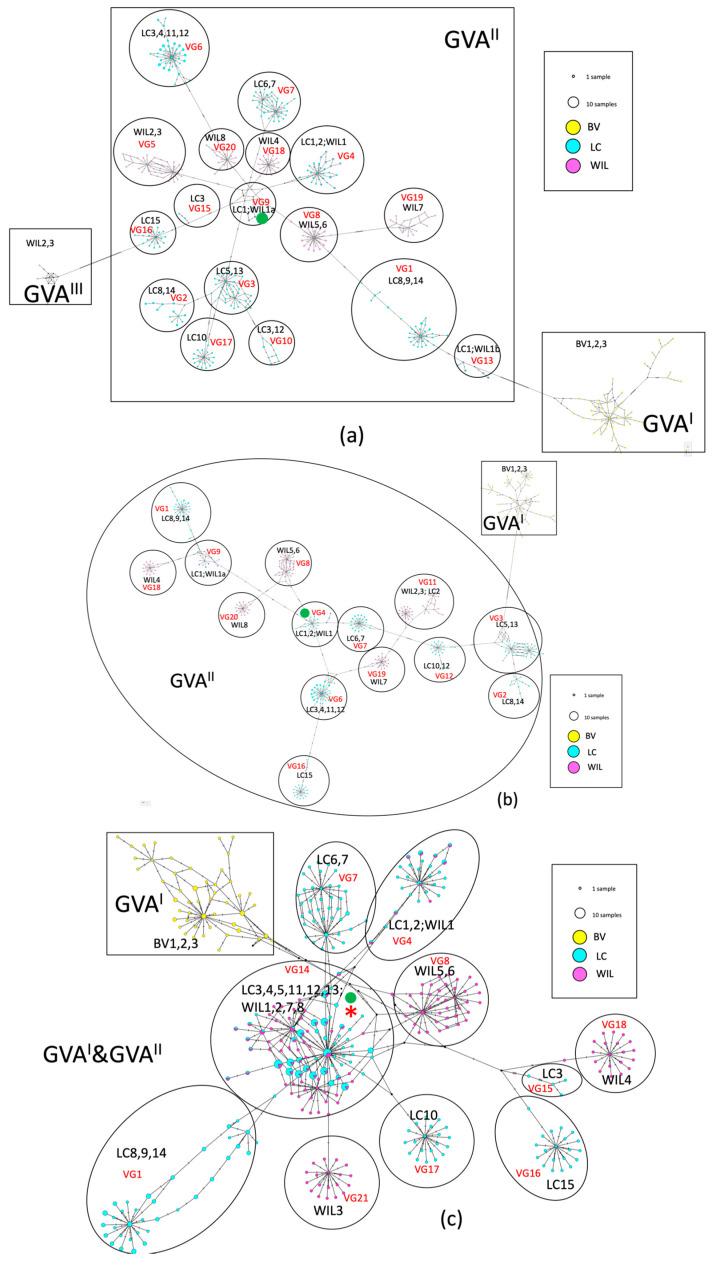
The intra- and inter- host diversity of coat protein (CP), movement protein (MP), and RNA-binding (RB) genes of grapevine virus A (GVA) within fifteen Shiraz disease (SD)-affected grapevines from Langhorne Creek (LC), eight SD-affected grapevines from Willunga (WIL), and three leafroll disease-affected grapevines from Barossa Valley (BV) were analyzed by phylogenetic median-joining haplotype networks (MJNs) clustering. (**a**–**c**) represent MJNs of the CP, MP, and RB genes, respectively. After amplicon sequence clustering, sequences that shared 100% nucleotide identity were merged into unique variants. The sequences of the 20 largest clusters of amplicon variants of each gene from each of the 26 grapevines from the three locations were used for analysis. Yellow, blue, and purple colors represent variants from BV, LC, and WIL, respectively. After MJNs were constructed, variants were grouped together to form variant groups. The original copy numbers of each variant are ignored. The size of the variant circles indicates the number of samples in which the identical variant was found. The hatch mark indicates the number of mutations between two variant clusters. The primary ancestral variant group of each gene is marked by a green dot in each subfigure. The supervariant group VG14 of the RB gene is labeled by a red asterisk in Figure 3. GVA variants from phylogenetic groups I, II, and III are labeled as GVA^I^, GVA^II^, and GVA^III^. GVA^I^ and GVA^II^ variants of the RB gene were together due to high similarity and labeled as GVA^I&II^. The number of hatch marks represents the number of mutations between each of the two variant clusters. The hypothetical median vectors (black dots) demonstrated the hypothetical connection of two variant clusters when connections are missing. See Appendix A for enlarged images of the GVA^I^ variants of Figure 2a–c.

**Figure 3 viruses-16-00042-f003:**
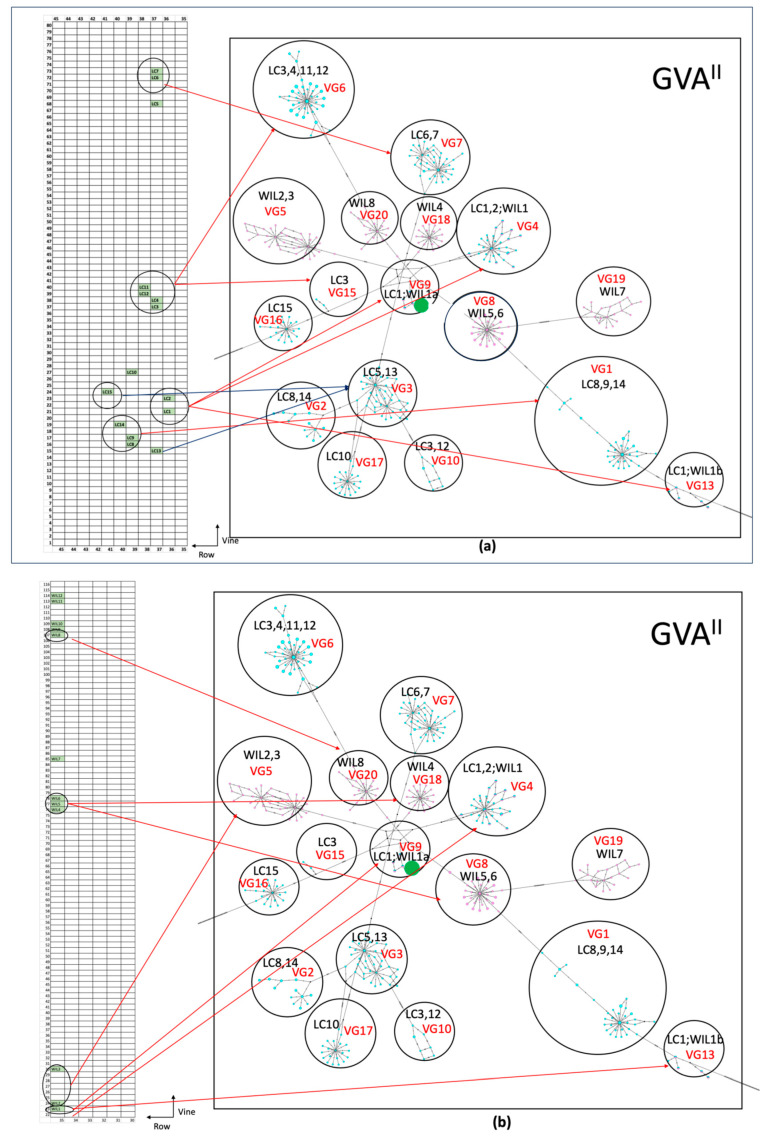
Median-joining haplotype networks of the coat protein (CP) gene versus relative geographical locations of grapevines from (**a**) the Willunga vineyard (WIL1 to 8) and (**b**) the Langhorne Creek vineyards (LC1 to 15). Relative geographical locations of each grapevine within the vineyard are shown on the left, with row and grapevine numbers indicated and sampled grapevines shaded in green. Red arrows point to the multiple- or single-sample variant groups identified previously. Variant group IDs and symbols in the MJN are detailed in Table 4 and Figure 2a.

**Table 1 viruses-16-00042-t001:** Overall number of reads and unique variants before and after clustering of grapevine virus A amplicon sequences across 26 grapevines.

Gene ^1^	MP	CP	RB
Total raw reads	6,466,112	8,784,286	7,568,478
Quality trimmed reads	2,805,669	3,877,170	3,399,212
Size trimmed reads	2,188,151	2,080,245	3,153,286
Primer trimmed reads	2,128,784	2,015,120	2,814,646
Proportion of trimmed reads in total raw reads (%)	64.44	46.08	73.36
Total nucleotide (nt) variants	12,983	13,103	10,052
Unique nt variants ^2^	10,454	10,899	6366
Proportion of unique nt variants in total variants (%)	80.52	83.18	63.33
Total amino acid (aa) variants	6593	5389	6192
Unique aa variants	4402	2250	1900
Proportion of unique aa variants in total variants (%)	66.77	41.75	30.68

^1^ MP = movement protein, CP = coat protein, RB = RNA binding protein. ^2^ Total number of unique nucleotide variants for each gene.

**Table 2 viruses-16-00042-t002:** Lowest percentage pairwise nucleotide identities and amino acid similarities of grapevine virus A (GVA) phylogenetic groups (GVA^I^, GVA^II^, and GVA^III^) amongst 26 grapevine samples.

Lowest Identity ^a^	MP ^b^		CP ^b^		RB ^b^	
	% nt Identity ^a^	% aa Similarity ^a^	% nt Identity	% aa Similarity	% nt Identity	% aa Similarity
GVA^I^ variants	95.60% ^1^	95.04% ^1^	92.59% ^1^	86.42% ^1^	90.40% ^3^	86.36% ^3^
GVA^II^ variants	88.74%	90.08%	86.42%	87.04%	See GVA^I^	
GVA^III^ variants	N/A		88.07% ^2^	88.89% ^2^	N/A	

^a^ nt = percentage nucleotide identity, aa = percentage amino acid similarity, GVA^I^, GVA^II^, GVA^III^, grapevine virus A isolates from the phylogroups I, II, and III, respectively. ^b^ MP = movement protein, CP = coat protein, RB = RNA binding protein ^1^ Identities or similarities calculated from samples BV1, BV2, and BV3. ^2^ Identities calculated from GVA^III^ variants from samples WIL2, WIL3, and WIL8. ^3^ According to the phylogenetic analysis of grapevine virus A [15], variants of GVA^I^ and GVA^II^ of the RB gene cannot be differentiated and, therefore, the sequences of all samples with these variants were combined.

**Table 3 viruses-16-00042-t003:** Sequence similarity of the largest 5 clusters to the two metagenomic high throughput sequencing isolates LC1-1 and LC1-2 that were obtained from the same grapevine LC1.

Gene ^1^	Sample ID	No. of Reads	% Nucleotide (nt) Identity to LC1-1 ^3^	% nt Identity to LC1-2 ^3^
CP	LC1_CP_Cluster1 #	9679	96.09%	100.00% #
	LC1_CP_Cluster2 #	8968	96.30%	99.79% #
	LC1_CP_Cluster3 *	7105	100.00% *	96.09%
	LC1_CP_Cluster4 #	4359	96.09%	99.59% #
	LC1_CP_Cluster5	1345	98.97%	97.12%
MP	LC1_MP_Cluster1 *	22,327	100.00% *	93.96%
	LC1_MP_Cluster2 *	16,980	99.73% *	94.23%
	LC1_MP_Cluster3 #	3972	93.96%	100.00% #
	LC1_MP_Cluster4 #	547	94.51%	99.45% #
	LC1_MP_Cluster5 *	377	99.18% *	94.78%
RB	LC1_RB_Cluster1 *	61,140	100.00% *	N/A ^2^
	LC1_RB_Cluster2	41,319	96.97%	N/A
	LC1_RB_Cluster3	8577	96.46%	N/A
	LC1_RB_Cluster4	2652	97.47%	N/A
	LC1_RB_Cluster5 *	2626	99.49% *	N/A

* Variants share more than 99.4% nt identities to LC1-1. # Variants share more than 99.4% nt identities to LC1-2. ^1^ MP = movement protein, CP = coat protein, RB = RNA binding protein. ^2^ Since isolate LC1-2 includes only partial sequences of the RB gene, sequence comparison was not performed for the RB gene and shown as N/A. ^3^ The isolate LC1-1 (accession no. MT070963) and LC1-2 (accession no. MT070962) were detected in the same grapevine LC1.

**Table 4 viruses-16-00042-t004:** Single-sample and multiple-sample variant subgroups identified for phylogroup II grapevine virus A (GVA) variants.

	Variant Group ID ^1^	CP ^2^		MP ^2^		RB ^2^	
		Sample ID	Lowest % Nucleotide (nt) Identity within the Variant Group	Sample ID	Lowest % nt Identity within the Variant Group	Sample ID	Lowest % nt Identity within the Variant Group
Multiple-Sample Variant Groups	VG1	LC8, 9, 14	95.68	LC8, 9, 14	97.53	LC8, 9, 14	95.96
	VG2	LC8, 14	98.77	LC8, 14	97.8	N/A ^3^	N/A
	VG3	LC5, 13	98.97	LC5, 13	97.53	N/A	N/A
	VG4	LC1,2; WIL1	98.56	LC1,2; WIL1	96.98	LC1,2; WIL1	95.96
	VG5	WIL2, 3	97.33	N/A	N/A	N/A	N/A
	VG6	LC3, 4, 11, 12	94.65	LC3, 4, 11, 12	96.98	N/A	N/A
	VG7	LC6, 7	98.97	LC6, 7	99.45	LC6, 7	97.98
	VG8	WIL5, 6	99.59	WIL5, 6	98.9	WIL5, 6	98.48
	VG9	LC1; WIL1a	99.38	LC1; WIL1a	98.63	N/A	N/A
	VG10	LC3, 12	98.56	N/A	N/A	N/A	N/A
	VG11	N/A	N/A	WIL2, 3; LC2	98.08	N/A	N/A
	VG12	N/A	N/A	LC10, 12	94.78	N/A	N/A
	VG13	LC1; WIL1b	99.38	N/A	N/A	N/A	N/A
	VG14 *	N/A	N/A	N/A	N/A	LC3,4,5,11,12,13; WIL1,2,7,8 *	97.47
Single-Sample Variant Group	VG15	LC3	99.38	N/A	N/A	LC3	97.47
	VG16	LC15	99.38	LC15	99.45	LC15	98.99
	VG17	LC10	99.59	N/A	N/A	LC10	98.99
	VG18	WIL4	99.59	WIL4	98.9	WIL4	98.99
	VG19	WIL7	98.97	WIL7	99.18	N/A	N/A
	VG20	WIL8	98.97	WIL8	99.18	N/A	N/A
	VG21	N/A	N/A	N/A	N/A	WIL3	98.99

^1^ Phylogroups contain phylogroup II GVA variants that were used to analyze the intra- and inter-host diversity in each of the 26 grapevines with either Shiraz disease or mild leafroll disease in this study (Figure 2). ^2^ MP = movement protein, CP = coat protein, RB = RNA binding protein. ^3^ Not applicable. * Super variant groups of the RB gene.

## Data Availability

The data presented in this study are available on request from the corresponding author. The data are not publicly available due to confidentiality.

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
