# Peer review of "Genetic Diversity of Grapevine Virus A in Three Australian Vineyards Using Amplicon High Throughput Sequencing (Amplicon-HTS)"

_viruses, 2023, doi:10.3390/v16010042_

Round 1
Reviewer 1 Report (Previous Reviewer 1)
Comments and Suggestions for Authors
The manuscript by Wu et al. deals with the genetic variability of grapevine virus A (GVA) strains infecting Syrah disease-affected grapevine plants.
This work is a resubmission of a previously evaluated manuscript. At that time, some observations were made, and there is no record of consideration of such observations.
The evaluation in general terms is consistent with the one presented for the manuscript viruses-2526336. The main concern is the lack of explanation about the low number of sequences remaining after filtering for the MP in the BV samples. As the authors stated, during the size filtering, some short sequences are GVA-related. It would be interesting to evaluate the occurrence of genuine deletions in the coding sequence, which could be reports of natural variants. The occurrence of triplets deletion among GVA strains was identified (see Goszczynski et al., 2012). The amplicon sequencing analysis provides enough information to support the significance of such deletions, allowing for clear discrimination between genuine deletions and sequencing errors.
The authors performed size filtering for each amplicon up to 4 nucleotides below the amplicon size. How was this threshold defined? Deletion of 1, 2, or 4 nucleotides could lead to a frame shift, with potential biological implications. Considering insertions or deletions of triplets might make more sense in this context.
The primers used for RB yield a fragment of 273 nt, but the sequences were filtered for 230-244 nucleotides. What is the reason for modifying the criteria used for the other ORFs?
Author Response
The evaluation in general terms is consistent with the one presented for the manuscript viruses-2526336. The main concern is the lack of explanation about the low number of sequences remaining after filtering for the MP in the BV samples.
Answers: We believe this issue is related to primer binding specificity. The MP primers were designed by utilizing complete genome sequences obtained from the GenBank database, aiming to cover all existing sequences worldwide. However, during that period, there were limited full-length GVA sequences available in GenBank. Moreover, the GVAI variants from BV had not been fully sequenced at that time. Consequently, the forward primers had 4 mismatches with the complete genome sequences from BV (accession no. OP752632). An explanation has been added to section 4.1
4.1. Challenges of GVAI amplification by the MP primers
The number of reads in the BV samples (BV1 to BV3) was significantly reduced post-filtering, as indicated in Table S3. In fact, the unfiltered reads from BV contain a substantially higher percentage of host-associated reads compared to samples from WIL and LC, which may indicate the binding specificity of the MP primer to GVAI variants was low. At the time of this study to specificity of the MP primers for detection of GVAI strains could not be determined as a sequence information was unavailable. Subsequent analysis has identified that there are four nt between the forward primer and the complete genome sequences of GVAI from BV (accession no. OP752632). The shorter GVA-associated sequences were likely a result of incomplete PCR amplification, where some amplicons were only partially amplified, or due to the occurrence of incomplete segments during the sequencing process.
As the authors stated, during the size filtering, some short sequences are GVA-related. It would be interesting to evaluate the occurrence of genuine deletions in the coding sequence, which could be reports of natural variants. The occurrence of triplets deletion among GVA strains was identified (see Goszczynski et al., 2012). The amplicon sequencing analysis provides enough information to support the significance of such deletions, allowing for clear discrimination between genuine deletions and sequencing errors.
Answers: The GVA-associated sequences we mentioned here are either incomplete segments generated by PCR or sequencing. A sentence of explanation was added to section 4.2.The method to identify insertions and deletions was added to section 2.4. See also the following:
“Amplicons of different sizes can arise due to incomplete extension, insertions, deletions and chimeras during amplification [72,73], leading to the inadvertent identification of artificial variants that affect downstream analyses. Therefore, untrimmed and trimmed sequences were filtered to the same size to ensure accuracy of the downstream phylogenetic and intra-host diversity analyses. Additionally, prior to further analysis, the trimmed sequences of the correct length were examined for presence of insertions and deletions, and none were identified.”
“Prior to clustering, an examination of insertions and deletions was conducted. Duplicates in primer-trimmed reads were eliminated from each sample using the “rmdup” command of SeqKit (v.2.3.0) [62]. Subsequently, the remaining reads were aligned using Muscle (v.3.8.31) [63]. The resulting aligned fasta files underwent visual assessment specifically focused on identifying insertions and deletions.”
The authors performed size filtering for each amplicon up to 4 nucleotides below the amplicon size. How was this threshold defined? Deletion of 1, 2, or 4 nucleotides could lead to a frame shift, with potential biological implications. Considering insertions or deletions of triplets might make more sense in this context.
The primers used for RB yield a fragment of 273 nt, but the sequences were filtered for 230-244 nucleotides. What is the reason for modifying the criteria used for the other ORFs?
Answers: We apologize for any confusion. The corrected amplicon size for the RB gene is 236 bp, and this correction has been reflected in Table S1. As previously noted, certain GVA variants, such as the isolate I327-5 (Goszczynski et al., 2014), may exhibit insertions and deletions in the overlapping region of ORF2 and ORF3. However, the amplicons we designed did not cover this overlapping region.
Our initial size criteria were designed to accommodate short insertions and deletions. However, upon reanalyzing the primer-trimmed amplicons through visual assessment of the alignment using Muscle, we did not observe any insertions or deletions. This was added the discussion section 4.2. The variations in sequence length were attributed to amplicons containing partial primer sequences, resulting in either shorter or longer sequences. The method we used for defining the size criteria is added to the method section 2.4.
“Amplicons of different sizes can arise due to incomplete extension, insertions, deletions and chimeras during amplification [72,73], leading to the inadvertent identification of artificial variants that affect downstream analyses. Therefore, untrimmed and trimmed sequences were filtered to the same size to ensure accuracy of the downstream phylogenetic and intra-host diversity analyses. Additionally, prior to further analysis, the trimmed sequences of the correct length were examined for presence of insertions and deletions, and none were identified.”
“The size criteria were determined based on a preliminary analysis of read length distribution of each gene region. Trimming and merging of reads were individually conducted for each sample using the "readlength" function of BBmaps. By considering all 26 samples, size criteria covering at least 80% of reads in the read-length histogram were established. This criteria differ for the MP, CP, and RB genes, adapting to the read length distribution of each gene region. This filtering process is implemented to exclude reads that are excessively long or short.”
“Prior to clustering, an examination of insertions and deletions was conducted. The output data did not use for clustering. Duplicates in primer-trimmed reads were eliminated from each sample using the 'rmdup' command of Seqkit (v.2.3.0)[62]. Subsequently, the remaining reads were aligned using Muscle (v.3.8.31) [63]. The resulting aligned fasta files underwent visual assessment specifically focused on identifying insertions and deletions.”

Reviewer 2 Report (New Reviewer)
Comments and Suggestions for Authors
Dear authors,
The manuscript on Genetic Diversity of Grapevine Virus A sounds good. But I have few points to be clarified as follows
- In the manuscript, the term “Variants” are used. Whether the variants differ in the pathogenicity characters, if so to what extent they differ is not mentioned
- Throughout in the text, the word “similarity” is used. Authors may kindly check and whether necessary change it to identify. Whether the diversity in nucleotides is reflected in amino acid changes- is not mentioned
- How much of the changes are observed in amino acid sequences in the coding region? Do they contribute to any modification in protein in functional motifs or modules?
- Line 137 – Remove “and” after Total Nucleic acid extraction.
- Line 124-125 – It is given as two RT-PCR with two sets of Primers (CP and RB), but in Line 140-141- It is given as amplification for three genes- MP, CP, and RB genes. This may be clarified and corrected.
- Line 156: Whether it is 1.8% gel (or) 1% gel. The amplicons are only less than 500 bp. Then why 1.8% gel.
- The exhaustive details of amplicon raw reads trimming and filtering (2.4) - Concise only to few lines, not as separate heading. Give only softwares used for trimming and filtering.
Dear authors,
Kindly check the sentence forming, moderate correction is needed.
Author Response
- In the manuscript, the term “Variants” are used. Whether the variants differ in the pathogenicity characters, if so to what extent they differ is not mentioned
Answers: GVA variants refer to genetic strains coexisting within a GVA population. Given that multiple variants often occur in mixed infections within a plant, it becomes challenging to separate and identify their individual pathogenic traits in field grapevines. Further work, such as using infectious clones, may be required to identify the pathogenic traits of these variants. A few sentences have been added to address this matter in sections 4.0 and 5.0. Please see also the yellow highlights in the following paragraphs:
Section 4.0
“This study used Amplicon-HTS of the MP, CP and RB genes of GVA to demonstrate the complexity of the genetic structure of a population of GVA variants in samples of individual grapevines expressing either SD or mild LRD symptoms in different vineyards. In this study, GVA variants refer to genetic strains that coexisted within a GVA population.”
Section 5.0
“Furthermore, delving into the pathogenic traits of individual GVA variants at both nucleotide and protein levels is essential. Investigating the correlation between nt and aa changes, and understanding how these aa changes contribute to modifications in protein functional motifs, necessitates the use of infectious clones to provide a comprehensive understanding of the association between each GVA variant and their role in the symptom expression of SD.”
- Throughout in the text, the word “similarity” is used. Authors may kindly check and whether necessary change it to identify. Whether the diversity in nucleotides is reflected in amino acid changes- is not mentioned
Answers: We use “identity” to denote nucleotide changes and “similarity” for amino acid changes. We have observed that both terms, “identity” and “similarity”, are used interchangeably for describing amino acid changes in published papers.
Regarding whether the diversity in nucleotides is reflected in amino acid changes, Table 1 shows that the number of unique aa variants was much less than nt variants after translation from nucleotides to amino acids. In Section 3.2, we mentioned that, after nucleotide sequence clustering, a total of 10,454, 10,899, and 6,366 unique variants were discovered from the MP, CP, and RB genes, respectively. When grouped again at 100% aa similarity, 4402, 2250, and 1900 distinct protein variants were identified for the MP, CP, and RB genes, respectively. The reduction in the number of variants after translation indicates that the majority of nt changes are synonymous substitutions. Two sentences have been added to section 4.2. Please see the yellow highlights below:
As depicted in Table 1, the translation of nt to aa sequences, followed by grouping at 100% aa similarity, revealed a notable reduction in the number of unique aa variants compared to nt variants. This observation suggests a predominant occurrence of synonymous substitutions among nucleotide changes. Although synonymous substitutions conventionally are thought to have no direct impact on protein structure or function, studies have demonstrated their role on changing virus fitness by altering the secondary structure of RNAs and modifying codon usage bias during replication [76-78]. These alterations may lead to a reduction in translation and replication efficiency, resulting in decreased infectivity during transmission and weakened protein expression during virus evolution [77-79]. The intricate interplay between synonymous and nonsynonymous mutations in the GVA population under natural selection pressure, and their implications for pathogenicity and the evolution of GVA, merits further exploration.
- How much of the changes are observed in amino acid sequences in the coding region? Do they contribute to any modification in protein in functional motifs or modules?
Answers: We have analyzed the maximum number of aa changes in each sample within the three gene regions, represented as “No. aa difference” in Table S4. Regarding the investigation of how aa changes contribute to alterations in protein functional motifs and modules, we acknowledge the absence of an effective protocol to demonstrate how amino acid changes specifically affect protein function. This requires the use of infectious clones and a significant amount of further analysis, which, although important, falls outside the scope of our paper. Our study primarily focused on genetic diversity at the nucleotide level within GVA population. The exploration of how amplicon-HTS can be utilized to investigate aa changes and conduct protein functional analysis should be a subject of investigation in the near future. The following sentences have been added to the end of section 5.0.
“Furthermore, delving into the pathogenic traits of individual GVA variants at both nucleotide and protein levels is essential. Investigating the correlation between nt and aa changes, and understanding how these aa changes contribute to modifications in protein functional motifs, necessitates the use of infectious clones to provide a comprehensive understanding of the association between each GVA variant and their role in the symptom expression of SD.”
- Line 137 – Remove “and” after Total Nucleic acid extraction.
Answers We removed “ and”.
- Line 124-125 – It is given as two RT-PCR with two sets of Primers (CP and RB), but in Line 140-141- It is given as amplification for three genes- MP, CP, and RB genes. This may be clarified and corrected.
Answers: The two sets of primers, CP and RB (Ah587/Ac995 and H7038/C7273), were used for virus detection purposes during sample selection. Table S1 provides information on the type of assay for each primer set. The CP primers used for amplicon HTS were longer than the ones used for virus detection, as listed in Table S1.
- Line 156: Whether it is 1.8% gel (or) 1% gel. The amplicons are only less than 500 bp. Then why 1.8% gel.
Answers: We prefer a high-percentage gel and a long running time to ensure the amplification of only a single-sized PCR product. 1.8% is the standard gel concentration in our lab.
- The exhaustive details of amplicon raw reads trimming and filtering (2.4) - Concise only to few lines, not as separate heading. Give only softwares used for trimming and filtering.
Answers: section 2.4 has been revised as follow:
2.4. Amplicon raw reads trimming and filtering
Raw reads with a quality score of under 20, and a length below 50 bp as well as the Illumina adapter sequences were removed using AdapterRemoval (v. 2.2.2) [58](quality trimming). The remaining reads from both directions were paired and merged using the “collapse” command of AdapterRemoval. Sequence reads were converted from “fastq” to the “fasta” format using the “reformat” function of BBmaps (v. 35.85) [59].
Amplicon sequence reads from the MP, CP, and RB genes were filtered to the designed size criteria: 399-404 nucleotide (nt) for MP , 560-564 nt, for CP, and 230-244 nt for RB using the “reformat” function of BBmaps (size trimming).The size criteria were determined based on a preliminary analysis of read length distribution of each gene region. Trimming and merging reads were individually conducted for each sample using the "readlength" function of BBmaps. By considering all 26 samples, size criteria covering at least 80% of reads in the read-length histogram were established. This criteria differ for the MP, CP, and RB genes, adapting to the read length distribution of each gene region. This filtering process is implemented to exclude reads that are excessively long or short.
Afterwards, the filtered amplicon sequence reads were compared against the local GVA sequence database out using the “blastn” command of BLAST+ to determine the orientation of GVA amplicon sequence reads and to filter out the unspecific reads. The local database of GVA genomes sequences was built using the ”makeblastdb” command using BLAST+ (v. 2.11.0) [60] as described by Wu et al. [15].
The Cutadapt (v. 3.4) software was then used to remove primer sequences from the amplicon sequence reads, implementing a mismatch-tolerant function for variations in primer binding sites using default parameters [61].
After primer trimming, reads were filtered again to the expected sizes of 364 bp, 524 bp, 198 bp for MP, CP, and RB genes, respectively. Additionally, the extra 38 nt sequence from the intergenic region right before the CP gene was deleted from the original CP amplicon sequences using the Cutadapt’s “cutadapt” command, resulting in a final amplicon size of 486 bp for the CP gene.
Prior to clustering, an examination of insertions and deletions was conducted. The output data did not use for clustering. Duplicates in primer-trimmed reads were eliminated from each sample using the 'rmdup' command of Seqkit (v.2.3.0)[62]. Subsequently, the remaining reads were aligned using Muscle (v.3.8.31) [63]. The resulting aligned fasta files underwent visual assessment specifically focused on identifying insertions and deletions.
Comments on the Quality of English Language
Dear authors,
Kindly check the sentence forming, moderate correction is needed.
Answers: Three of the coauthors are native English speakers and the manuscript has undergone a high level of scrutiny for language quality. One has re-checked the accuracy of the English language and found no quality issues.

Round 2
Reviewer 1 Report (Previous Reviewer 1)
Comments and Suggestions for Authors
The authors have answered the questions in the first review round. The manuscript has been improved from the original version. I consider it suitable for publication in Viruses in its present form.
This manuscript is a resubmission of an earlier submission. The following is a list of the peer review reports and author responses from that submission.
Round 1
Reviewer 1 Report
Comments and Suggestions for Authors
The manuscript of Hadidi et al. presents the genomic analysis of Grapevine Virus A (GVA) infecting grapevines through high-throughput sequencing (HTS) analysis of amplicons.
The implemented procedure appears to be an excellent tool for conducting comprehensive genomic analysis of viral populations
The detailed protocol and the interpretation of the results make this manuscript a valuable contribution, demonstrating the effectiveness of amplicon-HTS analysis as a powerful tool for conducting similar studies.
The main concern from my perspective regarding the manuscript is the filter process that was implemented.
The authors implemented a quality filter based on the size of the amplicons, restricting them to a specific size for each PCR product. As mentioned by the authors (Lines 508-512), insertions and deletions have a significant impact on the evolutionary process of viruses.
In particular, for the MP in BV samples, there is a remarkably low number of remaining reads after filtering. It would be interesting to analyze the filtered out sequences to investigate the occurrence of shorter or longer sequences resulting from genuine deletions or insertions. Such mutations are part of the evolutionary process experienced by the viral population in each plant and should not be neglected.
I believe that incorporating the analysis of reads carrying insertions and deletions will provide a more comprehensive snapshot of the actual viral population. By considering these variations, the analysis can capture a more accurate representation of the diversity and dynamics within the population.
The interpretation of the occurrence of different phylogenetic groups for each open reading frame (ORF) in the different samples leads to the conclusion that there are recombinant events taking place. However, the implications of these recombinant events are not explicitly mentioned in the manuscript. It would be valuable for the authors to discuss the potential significance and consequences of these recombination events in the context of viral evolution, spread, and potential impacts on host interactions.
Additionally, the confirmation of potential recombination events through long PCR, covering all three ORFs in a single fragment, could provide valuable information. This approach would allow for a more comprehensive analysis of the genomic structure and potential breakpoints, further supporting the presence of recombination events. It would be beneficial for the authors to consider incorporating such analyses to strengthen their findings and provide a more thorough understanding of the viral population dynamics.
Finally, this extensive analysis supports the taxonomic criteria currently adopted for the genus, which should be emphasized as it confirms this important taxonomic aspect. This finding reinforces the validity and relevance of the existing taxonomic classification and contributes to the broader understanding of the genus.
Reviewer 2 Report
Comments and Suggestions for Authors
The manuscript by Wu et al. is a study on the genetic diversity of GVA in Australia in vineyards affected by Shiraz Disease and Leafroll Disease. Authors use amplicon HTS and median-joining networks (MJN) on a total of 26 GVA infected samples to perform this study.
The aim of the study is interesting. The experimental work is well conducted and provides valuable data. However, I do not agree with the authors interpretation of the data. In my opinion, the main evolutionary conclusions are not supported by the data presented. In addition, the methodology used and the data obtained are not clearly presented. The manuscript is hard to follow. I recommend rejection and encourage resubmission considering the following issues:
1) I agree with the authors that the approach of using amplicon HTS in three genomic regions sounds interesting. Obtaining complete genomes by HTS, although more informative and complete approach, has a clear limitation on the number of sequences that can be obtained. On the other hand, cloning and Sanger sequencing of three genomic regions amplified by RT-PCR does not provide low frequency variants. However, with the amplicon HTS approach authors generate thousands of variants for each gene and then they use only a small amount of information. First, the genetic variability is only estimated by the two more distant sequences and they do not explore if this distance corresponds to few sequences very distant from the majority or a high variability between most of the variants. Second, they use these variants to perform MJN analysis, but they include only a small amount of the obtained variants, the most frequent ones. Which is then the point of obtaining thousands of variants including the low frequency ones if after that they only analyse 500? I would understand their approach if all the data were then used for the study. If this is not the case, I better trust in a different approach by recovering full genomes or at least partial sequences that they know correspond to the same isolate.
2) Authors use median-joining networks to infer evolutionary history. This approach, although visual and fast and easy to perform, has important limitations according to the literature (Kong, S. et al., 2016). In fact, MJN has been applied to viral studies cited by the authors (references 49-52) but using complete genomes, other phylogenetic analyses and/or following infection evolution. In my opinion, the snapshot authors present in this manuscript, that in addition correspond to a partial sequence and only for few of the variants obtained, can not support their evolutionary conclusions.
3) Discussion section (too long) should focus only in those aspects supported by the interpretation of the data. Although stating them, they should not emphasize on weak hyphotesis that are not supported by the data. For example, the identification of ancestors based only on a MJN partial snapshot and the conclusions on the viral transmission.
4) Authors should include only relevant data. Some data presented here is not used to support conclusions and do not contribute significantly to the study. It can be shown as supplementary. In addition, results should be presented more clearly, so that the paper is easy to follow and the reader do not get lost in a lot of data only related with variability of sequences.
I recommend exploring more deeply the variability data they have obtained, go further in diversity characterization of the most variable samples, for which they can obtain complete genomes, complete the study with other phylogenetic approaches and rewrite the manuscript accordingly and focusing on the relevant data to facilitate its reading.